# Global Neural Activities Changes under Human Inhibitory Control Using Translational Scenario

**DOI:** 10.3390/brainsci10090640

**Published:** 2020-09-16

**Authors:** Rupesh Kumar Chikara, Li-Wei Ko

**Affiliations:** 1Department of Biological Science and Technology, College of Biological Science and Technology, National Chiao Tung University, Hsinchu 300, Taiwan; rupesh.bt01g@g2.nctu.edu.tw; 2Center For Intelligent Drug Systems and Smart Bio-devices (IDS2B), National Chiao Tung University, Hsinchu 300, Taiwan; 3Institute of Bioinformatics and Systems Biology, National Chiao Tung University, Hsinchu 300, Taiwan; 4Brain Research Center, National Tsing Hua University, Hsinchu 30013, Taiwan; 5Drug Development and Value Creation Research Center, Kaohsiung Medical University, Kaohsiung 807, Taiwan

**Keywords:** electroencephalography (EEG), independent component analysis (ICA), event-related potential (ERP), N1, N2, P3, response inhibition, cingulate cortex, frontal cortex

## Abstract

This study presents a new approach to exploring human inhibition in a realistic scenario. In previous inhibition studies, the stimulus design of go/no-go task generally used a simple symbol for the go and stop signals. We can understand the neural activity of inhibition through simple symbol scenario. In the real world, situations of human inhibition are more complex than performing an experiment in the laboratory scale. How to explore the neural activities of inhibition in a realistic environment is more complex. Consequently, we designed a battlefield scenario to investigate the neural activities of inhibition in a more realistic environmental setting. The battlefield scenario provides stronger emotion, motivation and real-world experiences for participants during inhibition. In the battlefield scenario, the signs of fixation, go and stop were replaced by images of a sniper scope, a target and a non-target. The battlefield scenario is a shooting game between the enemy and the soldiers. In battlefield scenario participants played the role of the soldiers for shooting target and to stop shooting when a non-target appeared. Electroencephalography (EEG) signals from twenty participants were acquired and analyzed using independent component analysis (ICA) and dipole source localization method. The results of event-related potential (ERP) showed a significant modulation of the peaks N1, N2 and P3 in the frontal and cingulate cortices under inhibitory control. The partially overlapping ERP N2 and P3 waves were associated with inhibition in the frontal cortex. The ERP N2, N1 and P3 waves in the cingulate cortex are related to sustained attention, motivation, emotion and inhibitory control. In addition, the event-related spectral perturbation (ERSP) results shows that the powers of the delta and theta bands increased significantly in the frontal and cingulate cortices under human inhibitory control. The EEG-ERP waves and power spectra in the frontal and cingulate cortices were found more increased than in the parietal, occipital, left and right motor cortices after successful stop. These findings provide new insights to understand the global neural activities changes during human inhibitory control with realistic environmental scenario.

## 1. Introduction

In recent years, there has been a dramatic proliferation of research related to human response inhibition. The ability to suppress an ongoing motor action is known as human response inhibition that is necessary for the control of executive function. Generally, neural activities of human response inhibition are investigated using a stop-signal task or go/no-go task. The stop-signal task provides a way to test the problems of daily life activities, such as decision making during the time-sensitive condition. For example, stop while driving the car when the traffic light changes from yellow to red light [1,2,3,4,5,6]. Usually, past response inhibition studies used the simple symbol as go and stop signals [7,8]. These studies did not show a realistic environment for the subject when performing the stop-signal task. We can understand the neural activities of human inhibition. However, in the real world, investigating the neural activities of inhibition is more complex than on a laboratory scale. In order to identify the neural activities of human inhibition in a more realistic environment, our study designs a new experimental scenario to observe the neural activities of human response inhibition in a more realistic environment.

We used a battlefield scenario that is based on the traditional stop-signal task. This battlefield scenario incorporates a game that mimics gun fighting between soldier and enemy. In real-world situations, soldier should make correct decisions during critical points in a shooting, such as identifying potential targets as civilian or enemy. This battlefield scenario provides stronger motivation and real-world experience to participants than in the previous study of inhibition [7,8]. Previous studies of electroencephalography (EEG) and functional magnetic resonance imaging (fMRI) reported that frontal cortex and pre-supplementary motor area (pre-SMA) of the brain have been reported under human inhibition [9,10]. In addition, former studies of fMRI reported that the anterior cingulate cortex (ACC), the inferior parietal lobe (IPL), the pre-supplementary motor area (pre-SMA) and the ventrolateral prefrontal cortex (VLPFC) have been related to human response inhibition [7,8,11,12,13,14,15,16].

The high temporal resolution electroencephalography (EEG) methods were used including event-related potential (ERP) and event-related spectral perturbation (ERSP) to investigate the neural activities of human inhibition under battlefield scenario. The EEG has been widely used to investigate neural activities on a millisecond scale [17]. Previous studies used the EEG-ERP method to examine the brain activity under human inhibitory control in the frontal area of the brain [18,19,20]. The ERP-N2 wave is a negative potential evoked response in the frontal cortex of the brain. The ERP-N2 wave observed at 200 ms after response inhibition (stop stimulus) [21]. The N2 wave in the frontal cortex has been related to human inhibitory control [21,22,23]. In addition, the ERP P3 wave is a positive potential evoked response in the frontal cortex of the brain. The P3 wave appeared approximately 300–400 ms after stop stimulus and it has been associated with the response inhibition [24,25]. Furthermore, the delta (1–4 Hz) and theta (4–7 Hz) band powers were observed increase in the frontal cortex (200–600 ms) under response inhibition [26].

In stop-signal task, stop success and stop failure depend on a subject’s attention. If the subject is more attentive during the experiment they will be more likely to response to a stop signal successfully. Otherwise, they will be more likely to fail [21]. The functional role of cingulate cortex has been observed in cognitive control (inhibitory control), attention control and motor control [27]. Moreover, previous clinical studies reported that cingulate cortex has an important role in neurological disorders, like schizophrenia and depression [28,29]. The frontal cortex has been related to human inhibition and visual attention. However, some study reported that frontal cortex has been shown to be a crucial area for maintaining visual attention. Consequently, the frontal cortex has been considered as an important brain area for maintaining visual attention instead of stopping the action [30,31,32]. The parietal lobe has been related to the perception of emotions. The parietal lobe is generally considered associated with the visual stimuli that are less precisely related to inhibition [30]. The parietal lobe plays a functional role in the integration of sensory information from several areas of the brain. It also plays a role in the processing of information associated with the sense of touch [33]. The parietal lobe has been involved in visuospatial processing [34]. The occipital lobe has been related to visual stimuli [30,31,32]. The left and right motor cortices have been related to hand movement of the subjects [30,31,32].

In this study, we acquired electroencephalography (EEG) signals from twenty subjects and analyzed them using event-related potential (ERP) with independent component analysis (ICA) and dipole sources location. Total number of participants was greater than a previous study of inhibition [30]. In addition, to verify that the designed battlefield scenario was valid for a large number of samples, each participant performed 200 trials (200 × 20 = 4000 samples) more than in a previous study of human inhibition [30]. In this study, the subjects were sitting in a chair to perform the experiment. However, in the previous simultaneous functional magnetic resonance imaging (fMRI) and EEG study, each subject was lying positioned in fMRI scanner while observing a projected interactive environment using a mirror mounted on the head coil over the subject [30].

In this study, changes in brain activity were investigated in the frontal cortex, cingulate cortex, parietal lobe, occipital lobe, left motor cortex and right motor cortex of the brain (i.e., global neural activities changes) during human inhibitory control using a battlefield scenario. The aim of this study was to investigate the global brain activity changes during response inhibition in a realistic environmental setting. We assumed that ERP biomarker is observed in the cingulate cortex of the brain under inhibition of the human response. In this work, we used ERP analysis using independent component analysis and dipole sources’ locations to explore the neural activities under human inhibitory control in a realistic battlefield scenario.

## 2. Materials and Methods

### 2.1. Participants

Twenty right-handed healthy volunteers participated in the experiment (14 males and 6 females, mean age ± SD: 23.0 ± 0.9 years old). According to the self-reported inventory, all subjects had normal vision and none of them had history of neurological and psychiatric disorders. The study was approved by the Research Ethics Committee of the National Taiwan University, Taipei, Taiwan, NTU-REC No: 201210HS007. This study was carried out in accordance with the recommendations of the Institutional Review Board (IRB) of the National Taiwan University, Taipei, Taiwan. All subjects gave written informed consent by the laws of the country and the Research Ethics Committee of the National Taiwan University, Taipei, Taiwan.

### 2.2. Experimental Design

To investigate the response inhibition in a battlefield scenario, each participant was asked to perform a stop-signal task. The battlefield scenario was a modified stop-signal task, in which the fixation, go and stop stimuli were replaced with the images of sniper scope, target and non-target, respectively. The designed battlefield scenario mimicked a shootout between the enemy and the soldier, as shown in Figure 1. In the battlefield scenario, the participants played the role of soldiers—shooting the enemy and stopping shooting when a hostage was presented. The battlefield scenario provided participants with strong motivation and real-world experiences for the human inhibitory control.

In the battlefield scenario—as in a prior study of human inhibition—each subject performed go trials (75%) and stop trials (25%) [8,9,35]. In the go trials, participants were instructed to respond with a left-key click of a mouse within 1000 ms as they saw a target picture (go stimulus) after sniper scope picture (fixation). In the stop trial, each subject held their response (stop stimulus), when a non-target picture appeared; this was presented after the go stimulus. The difficulty of the stop-signal task was systematically manipulated by dynamically changing the time delay between the go and stop signals. This is called stop-signal delay (SSD) in the stop trial. The SSD was initially set at 150 ms. If participants successful stopped the response when a non-target picture appeared, the SSD of the next stop-trial would increase by 50 ms to make the next stop-trial more difficult. If participants failed to inhibit their action, then SSD of the next stop trial would decrease by 50 ms. The SSD was decreased and increased according to the subject’s performance with a staircase-tracking procedure.

### 2.3. Acquisition and Preprocessing of EEG Signals

The EEG signals were recorded with a sampling rate of 500 Hz with a 32-channel NuAmps EEG cap (BioLink, Ltd., Australia). The 32 EEG-channels were fixed according to the standard of 10–20 international system. The acquired EEG signals were first inspected to remove the noisy EEG signals, such as eye blinking, indoor power noise and muscle artifacts. After this, the downsample rate was reduced from 1000 to 250 Hz. A high-pass filter with a cutoff frequency of 1 Hz and a low-pass filter with a cutoff frequency of 50 Hz was applied to remove muscular artifacts and line noises. In addition, independent component analysis (ICA) was used to remove the various kinds of muscle artifacts from the EEG signals [36]. Figure 2 shows the EEG recording system used in the present study. In this study, each subject performed 200 trials with 75% go trials and 25% stop trials. The acquired EEG signals were segmented into a set of the epoch from −200 ms to 1000 ms after the stimulus. According to the experimental design, three groups of epochs were extracted that include successful go trial, successful stop trial and unsuccessful stop trial. The 0 ms refers to the onset of the “go” stimulus in the go trials as well as in the stop trials. In the stop trial, each participant inhibits their response, when appeared a non-target picture, it was presented after the go stimulus.

### 2.4. Independent Component Analysis (ICA)

The independent component analysis (ICA) is a commonly used statistical technique to find out the linear projections of the data that can maximize the mutual independences of estimated components. It has been demonstrated as an effective technique to solve blind source separation (BSS) problem [17,36]. In this study, the ICA was used to separate the independent brain activities from artificial noises, such as eye movements, muscle and heart activities. These artifacts were usually contaminated EEG signals. After obtaining the independent components, DIPFIT2 routines (a plug-in in EEGLAB) were applied to find the 3D locations of an equivalent dipole or dipoles based on a four-shell spherical head model [17]. Then, the independent components obtained from all subjects with similar scalp topographies, dipole locations and power spectra were clustered in the same group. Since the anterior brain regions were correlated to the response inhibition function, such as, prefrontal cortex [37], frontal cortex [38] and right inferior frontal gyrus, pre-supplementary motor area (pre-SMA) [39]. In the present study, the frontal, central, parietal, occipital, left motor and right motor cortices of the brain were found activated under inhibition. Therefore, these six brain regions were used for further analysis, like the event-related potential (ERP) and event-related spectral perturbation (ERSP) to observe the global neural activities changes under inhibition.

### 2.5. Behavioral Analysis

Behavioral analysis was measured in successful go, successful stop and unsuccessful stop conditions including the go reaction time (Go-RT), stop-signal delay (SSD), stop-signal reaction time (SSRT), successful go (SG) ratio and successful stop (SS) ratio. The Go-RT was estimated for each participant response in successful go trials. The SSD represents the time of each subject to inhibit the response after stop signal. Accordingly, we observed the SSD of each subject in the stop trials under battlefield scenario. The time of the stop process is defined by stop-signal reaction time (SSRT) for each subject in the stop-trials. Furthermore, the SSRT based on the horse-race model of stopping was investigated to characterize the inhibitory control ability [3].

### 2.6. Event-Related Spectral Perturbation (ERSP) Analysis

Event-related spectral perturbation (ERSP) analysis was used to measure the mean event-related changes in spectral power over time and frequencies. In ERSP analysis, we measured increase decrease power spectrum change in EEG signals during successful go, successful stop and unsuccessful stop conditions [17]. We used the *newtimef () function* in the EEGLAB toolbox to compare the difference between successful stop vs. unsuccessful stop trails, see the link for a detailed description of ERSP analysis (https://sccn.ucsd.edu/wiki/Chapter_11:_Time/Frequency_decomposition). The color at each image pixel shows power in (dB) at given frequency and latency relative to the time locking event. Usually, for *n* trials, if $f_{k}\left(f,t\right)$, is the spectral estimate of trial *k* at frequency *f* and time *t*.
(1)$$ERSP\left(f,t\right)=\frac{1}{n}\overset{n}{\underset{k=1}{\sum }}{\left|f_{k}\left(f,t\right)\right|}^{2}$$

### 2.7. Statistical Analysis

In the EEG analysis, the post-stimulation effects in the human brain during successful go, successful stop and unsuccessful stop trials were investigated by transforming the EEG data after each epoch into the time and frequency domain using the ERSP and ERP methods [17]. In the ERP analysis, the Wilcoxon signed rank test shows the significant (FDR-adjusted *p* value < 0.01) difference between successful stop versus unsuccessful stop in red asterisk. The sky blue asterisk shows the significant difference between successful go versus successful stop. The yellow asterisk displays the significant difference between successful go versus unsuccessful stop. Moreover, in the ERSP analysis, the statistically significant differences in successful go, successful stop and unsuccessful stop trials were evaluated using the bootstrap method [17,40] with the significance threshold at *p* value < 0.05. The average ERP and ERSP analysis was investigated at each independent component cluster including frontal, central, parietal, occipital, left motor and right motor cortices in EEGLAB with Matlab [17,41].

## 3. Results

### 3.1. Behavioral Results

The behavioral performance of participants was examined during inhibitory control in the battlefield scenario. Subsequently the stopping mechanism itself cannot be directly observed, the SSRT was measured by subtracting the SSD from the Go-RT, such as (SSRT = Go-RT − SSD). In addition, the inhibition function (SS ratio) was computed as the number of SS trials divided by the number of all stop trials including successful stop and unsuccessful stop trials. The SG ratio was observed as the number of SG trials divided by the number of all go trials. Moreover, in battlefield scenario, the average Go-RT was observed 368 ± 74 ms. The SSD was investigated 194 ± 44 ms and SSRT was observed 174 ± 31 ms. The SG ratio was measured 92.0 ± 10% and SS ratio was observed 50.3 ± 3.25%, respectively.

### 3.2. Electroencephalography (EEG) Results

#### 3.2.1. Independent Component Scalp Map and Location of Dipole Sources

In the EEG analysis—after doing ICA process—the independent components were analyzed using DIPFIT2 routines—an EEGLAB plug-in—to find the 3D location of an equivalent dipole based on a four-shell spherical head model. Among independent components from twenty subjects, those with similar scalp topographies, dipole locations and power spectra were clustered together using K-means clustering toolbox (K = 6) in EEGLAB. The six component clusters were selected from the twenty subjects with similar topographic maps. Figure 3A–F displays the scalp maps and the corresponding equivalent dipole location of six clusters including frontal cortex (14Ss, 14ICs), cingulate cortex (13Ss, 13ICs), parietal cortex (14Ss, 14ICs), occipital cortex (16Ss, 16ICs), left motor cortex (15Ss, 15ICs) and right motor cortex (15Ss, 15ICs). We denote subjects (Ss) and independent components (ICs) in each cluster, respectively. For a more detailed description, see the link related to average and single dipole source location analysis (https://sccn.ucsd.edu/eeglab/dipfittut/dipfit.htmlold).

#### 3.2.2. Event-Related Potentials (ERP) under Human Inhibition

Figure 4A–F shows the average ERP waves of the frontal, cingulate, parietal, occipital, left motor and right motor cortices under target (go cue stimuli) and non-target (stop-signal stimuli) condition in successful go, successful stop and unsuccessful stop trials. The traces in bold colors marked on the ERP waves indicate the significant differences of the potentials exist between the successful go and successful stop trials (blue trace) and between the successful go and unsuccessful stop trials (yellow trace) with Wilcoxon signed rank test (*p* < 0.01). The red asterisks indicate where significant differences of the ERP exist between the successful stop and unsuccessful stop trials with Wilcoxon signed rank test (*p* < 0.01). In this study, on the upper side of the *y*-axis, we have indicated the negative (N) amplitude (−) from 0 to −6 µV and lower side of the *y*-axis, we have showed the positive (*p*) amplitude (+) from 0 to 6 µV.

Figure 4A,B shows the significant N2 negative wave around 400 ms after non-target (stop signal) 80 ms faster in the frontal and cingulate cortices under successful stop than in unsuccessful stop and go trials. In addition, we observed a significant P3 positive wave around 500 ms after non-targets (stop signal) 50 ms earlier in the frontal and cingulate cortices of the brain under successful stop than in unsuccessful stop and go trials. These findings show that N2 and P3 waves partially overlap in the frontal cingulate cortices are the neural markers of human inhibition. The significant N1 negative peak around 300 ms after non-target stimuli was investigated 40 ms faster in frontal and cingulate cortices of the brain under successful stop than in unsuccessful stop and go trials. The ERP N1 wave in the frontal and cingulate cortices is related to sustained attention, motivation, emotion. These results show that all participants were more focused on the non-target stimulus under the battlefield scenario.

Figure 4C shows the significant N2 negative peak around 400 ms and the P1 positive peak around 300 ms were elicited by the non-target stimuli in the parietal lobe under successful stop than in unsuccessful stop and go trials. Figure 4D displays the significant N2 negative peak around 400 ms and the P1, P3 positive peaks around 300 ms–500 ms were elicited by the non-target stimuli in the occipital lobe under successful stop than in unsuccessful stop and go trials. The ERP P1 wave in the parietal and occipital cortices is related to visual stimulation. The parietal and occipital lobes have played the function role in visual stimulation.

Moreover, Figure 4E shows the small N2 and P3 waves after the non-target (stop signal) stimuli in the left motor cortex under successful stop than in unsuccessful stop and go trials. This finding shows that the left motor cortex played a role in the movement of the hand. However, in the right motor cortex, we found no changes in EEG activity under response inhibition, as shown in Figure 4F. We observe the small ERP wave’s changes in the parietal cortex and the motor cortex. These brain regions are less effective with inhibitory control.

#### 3.2.3. Event-Related Spectral Perturbation (ERSP) during Human Inhibition

Figure 5A–D displays the average ERSP changes occurring in the frontal cortex, cingulate cortex and parietal cortex, the red color showing significant (*p* < 0.05) power changes in successful go (A), successful stop (B), unsuccessful stop (C) and successful versus unsuccessful (D) conditions. Figure 5A shows the beta band power (13–30 Hz) was observed significantly increase from 700 ms to 1000 ms after the target (go cue) stimuli in successful go (SG) condition at frontal and cingulate cortices. This finding is related to the hand movement of the subjects. Figure 5D displays the delta (1–4 Hz), theta (4–7 Hz) band powers were significantly increased approximately 200 ms to 400 ms elicited by non-target (stop signal) stimuli in successful stop (SS) vs. unsuccessful stop (US) condition at frontal and cingulate cortices. These finding suggests that frontal and cingulate brain areas are highly related to the human response inhibition.

Figure 5A–C shows that the powers of the delta (1–4 Hz), theta (4–7 Hz) bands increased significantly, and the powers of the alpha–beta bands decreased in the conditions of successful go (SG), successful stop (SS) and unsuccessful stop (US) at the parietal cortex. We observed that the powers of the alpha (8–12 Hz) and beta (13–30 Hz) bands decrease under the conditions of successful stop versus unsuccessful stop (SS-US) in the parietal cortex, as shown in Figure 5D. These EEG results are related to the movement of the subjects’ hands.

Figure 6A–C shows the delta (1–4 Hz), theta (4–7 Hz) band powers were observed significantly increase in successful go, successful stop and unsuccessful stop conditions at the occipital, left motor and right motor cortices. In addition, we investigated alpha (8–12 Hz) and beta (13–30 Hz) band powers decreased in successful go, successful stop and unsuccessful stop conditions at the occipital, left motor and right motor cortices, as shown in Figure 6A–D. These EEG results are related to the movement of the subjects’ hands.

The average ERSP changes that observed in the frontal, cingulate and left motor cortices show an increase in theta power (4–7 Hz) significantly around 200 ms to 650 ms and a beta power (13–30 Hz) around 800 ms to 1000 ms after the non-target stimulus in unsuccessful stop trials. These findings suggest that stimulation of theta and beta powers occurred due to failed stop (i.e., acute stress) in the frontal, cingulate and left motor cortices.

## 4. Discussion

In this study, neural activities of human inhibitory control were investigated under battlefield scenario. Our study provided real-world experience of open firing in public places between enemy and soldiers. In this work, we investigated the global neural activity changes under human inhibitory control in the frontal, cingulate, parietal, occipital, left motor and right motor cortices. The N2 and P3 waves were observed in cingulate, parietal, occipital and left motor cortices of the brain after successful stop. Additionally, the P1 wave was occurred in parietal and occipital cortices of the brain. These EEG results show the EEG biomarkers in the cingulate, parietal, occipital, left motor and right motor cortices of the brain under human inhibitory control with real-world scenario. The ERP-N1 wave was observed in frontal and cingulate cortices of the brain under inhibition. The ERP-P1 wave was examined in parietal and occipital cortices during response inhibition. These results are related to visual attention of the subjects.

The previous studies observed that response inhibition is a cognitive function of executive control [3,42,43,44,45,46]. The preceding behavior studies reported that SSD represents the time in which it is possible for a subject to inhibit their response. The probability of the successful response inhibition has been related to the subject’s attention level [47]. Accordingly, in the behavioral analysis we measured SSD, SSRT and RT for each participant under the battlefield scenario. The preceding neuroimaging studies have demonstrated that more activated brain regions of response inhibition under stop-signal task, such as ventrolateral prefrontal cortex, pre-supplementary motor area, prefrontal cortex, frontal cortex and right inferior frontal gyrus [37,38,39,48]. These previous investigations have reported that the frontal cortex is highly related to the human response inhibition. In present study, we found similar inhibition-related EEG-ERP markers of N2 and P3 waves, and the ERSP of delta–theta band powers increased in frontal cortex of the brain. Moreover, we explored N2 and P3 waves in cingulate, parietal, occipital and left motor cortices of the brain after successful stop. Furthermore, in clinical research, human inhibition deficits have been related to psychological disorders, such as attention deficit hyperactivity disorder [49,50,51], obsessive compulsive disorders [52].

### 4.1. EEG-ERP N2 and P3 Waves under Human Inhibitory Control

In present work, we investigated ERP-biomarker of human response inhibition under battlefield scenario. The EEG-ERP neural activity changes were found in the frontal, central, parietal, occipital, left motor and right motor cortices of the brain during inhibition. The inhibition-related N2 and P3 waves were identified in frontal and central cortices of the brain, as shown in (Figure 4A–F). The N2 wave means, the negative evoked response with its peak around 400 ms after non-target stimuli. The P3 wave means, the positive evoked response with its peak around 500 ms after non-target stimuli. The previous ERP studies reported that N2 and P3 waves have been associated with human response inhibition in the frontal cortex [21,22,53]. The ERP-N2 and P3 wave’s results in present study are consistent with former studies of response inhibition in only frontal cortex [21,22,53]. These findings suggest that frontal brain area is highly responsive to the human inhibitory control. Moreover, in our study, the ERP results of the N2 and P3 waves in the frontal cortex have been replicated according to previous studies of human inhibition [54]. In present work, the ERP-N2 wave modulation was observed higher in successful stop than in unsuccessful stop condition. Furthermore, a higher N2 wave amplitude in successful stop condition has already been found in the previous studies of the human inhibition [21,22]. In addition, the ERP-P3 wave has been observed in frontal cortex of brain, these outcomes completely consistent with previous study of response inhibition [53]. The cingulate cortex has been involved with emotion, motivation, learning and memory. It also plays a role in executive function, such as attentional control [27,55]. In this study, we measured the ERP-N2 and P3 waves neural markers of inhibition in the cingulate cortex of the brain after successful stop. In addition, the N1 wave was observed in the frontal cortex of the brain this finding is related to attention during response inhibition.

### 4.2. EEG-Power Spectral Changes under Inhibitory Control

The response inhibition-related delta and theta band powers were found to increase in the frontal and cingulate cortices of the brain under successful stops, as shown in (Figure 5B,D). These ERSP results show that the frontal and cingulate cortices are related to response inhibition. The delta and theta band power increases in frontal cortex are consistent with previous studies of response inhibition [56,57,58]. Moreover, beta power was found to increase in the frontal cortex of the brain during successful go tests, as shown in (Figure 5A), this ERSP result is related to the hand movement of the subjects [59,60,61]. Furthermore, the delta–theta band powers were found to increase in the cingulate cortex of the brain during the successful stop condition, as shown in (Figure 5D). This result shows that cingulate cortex is also responsive for movement of the hand and inhibition. In addition, previous studies report that the theta band power increase in the frontal cortex is associated with attention during response inhibition [62,63]. However, the alpha and beta band powers were found to decrease in the parietal, occipital, left motor and right motor cortices of the brain. These findings are related to hand response in the successful go condition.

In a previous study of simultaneous fMRI and EEG, the investigators observed changes in local brain activity simply in the pre-supplementary motor area, the right middle frontal gyrus and the left and right middle occipital gyrus [30]. However, in the present work, we found changes in global neural activity under human inhibitory control in the frontal, cingulate, parietal, occipital, left motor and right motor cortices. In the current study, the ERSP delta–theta band powers were found to increase during response inhibition in the frontal cortex. This result is similar to the previous study of inhibition [30]. Furthermore, the powers of the delta–theta band increased in the cingulate cortex. This finding is related to sustained attention, emotion, motivation and inhibitory control. These ERSP results are inconsistent with the previous inhibition study [30]. The ERP N1 wave in the frontal and cingulate cortices is related to sustained attention in a realistic environmental setting. This result is different from previous inhibition studies [30,54]. Therefore, the present work provides a new approach to understand changes in global neural activities of response inhibition.

## 5. Conclusions

The present study introduced a new experimental design of battlefield scenarios to explore the global neural activities of human inhibitory control in a realistic environment. We observed s partial overlap of ERP-N2 and P3 waves neural markers of human inhibitory control in the cingulate cortex of the brain in the battlefield scenario. Traditionally, the cingulate cortex has been related to positive emotion, motivation, inhibitory control and attentional control. In addition, we observed inhibition-related N2-P3 waves and increased delta–theta band powers in the frontal, parietal, occipital, left motor cortices of the brain under battlefield scenario. However, we found the small ERP waves changes in the parietal cortex and the motor cortex. These brain regions are less effective with inhibitory control. Present study explored the global neural activities of human inhibitory control under realistic scenario. These findings can be utilized to improve the psychopathology of obsessive-compulsive disorder (OCD), attention deficit hyperactivity disorder (ADHD) and schizophrenia.

## Figures and Tables

**Figure 1 brainsci-10-00640-f001:**
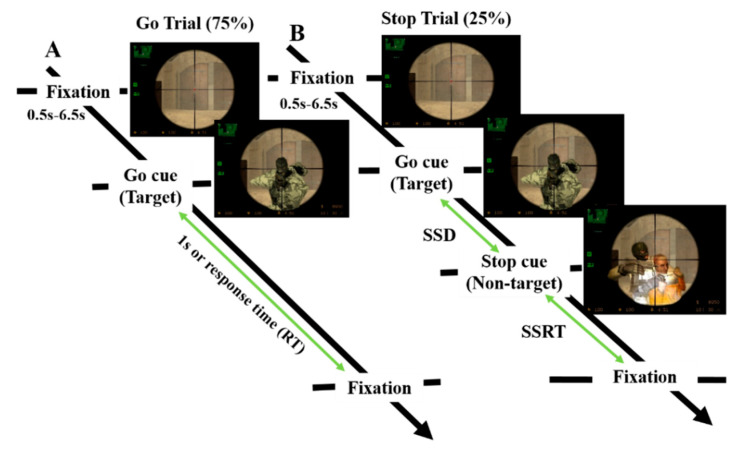
Experiment design of the battlefield scenario. (**A**) In the go trial, the subject was instructed to press the left button of a mouse in response to a go cue (target) stimulus as quickly as possible; (**B**) In the stop trial, a stop-signal (non-target) stimulus was used to inhibit the response. We measured each subject stop-signal delay (SSD) and stop-signal reaction time (SSRT) in the stop trials and response time (RT) in the go trials.

**Figure 2 brainsci-10-00640-f002:**
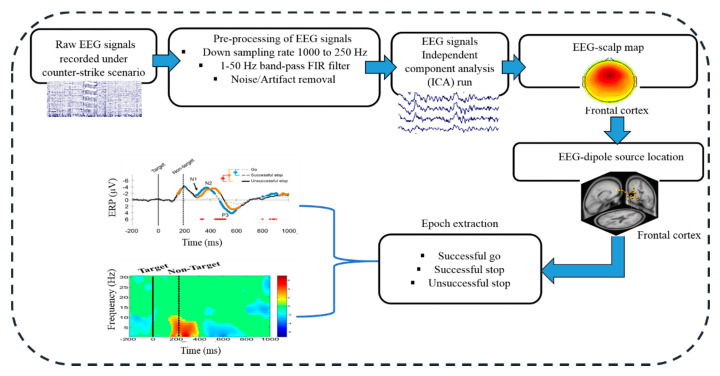
Flowchart of EEG signal analysis during human inhibition with counter-strike scenario.

**Figure 3 brainsci-10-00640-f003:**
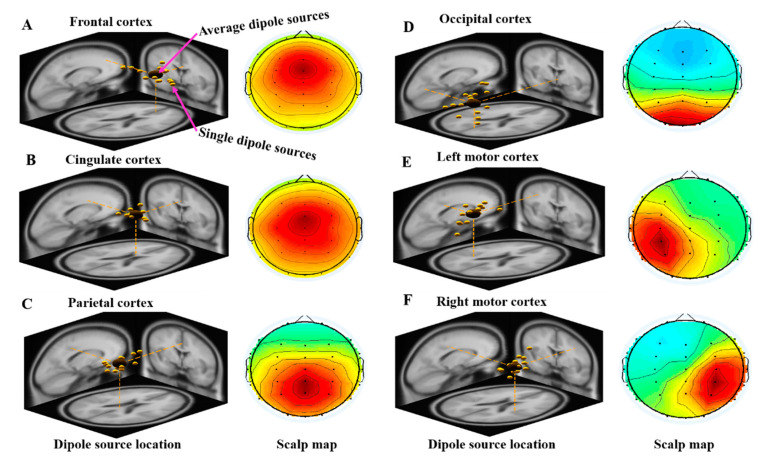
EEG dipole-source location and scalp map of six clusters including (**A**) frontal cortex (14Ss, 14ICs), (**B**) cingulate cortex (13Ss, 13ICs), (**C**) parietal cortex (14Ss, 14ICs), (**D**) occipital cortex (16Ss, 16ICs), (**E**) left motor cortex (15Ss, 15ICs) and (**F**) right motor cortex (15Ss, 15ICs). We represent subjects (Ss) and independent components (ICs) in each cluster. Yellow dots indicates the individual dipole source location of each subject; the large dark dot indicates the average dipole source location of all subjects.

**Figure 4 brainsci-10-00640-f004:**
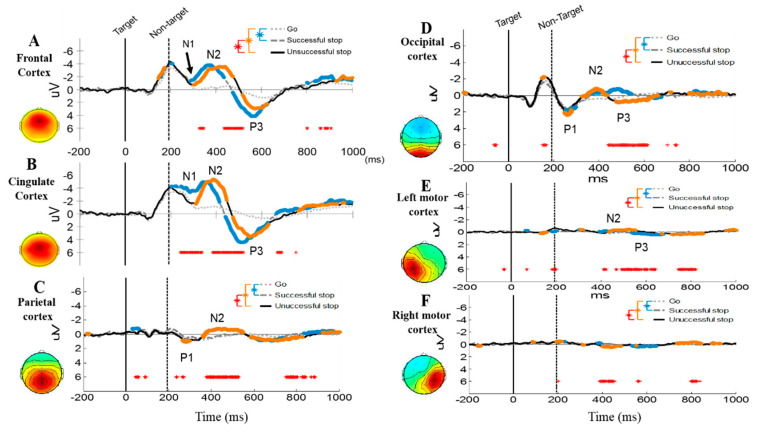
Averaged ERP of all subjects in the frontal cortex (**A**), cingulate cortex (**B**), parietal cortex (**C**), occipital cortex (**D**), left motor cortex (**E**) and right motor cortex (**F**) during successful go, successful stop and unsuccessful stop conditions with battlefield scenario. The ERP (µV) reveals on *y*-axis and time value of −200 to 1000 millisecond (ms) shows on *x*-axis. The black vertical line shows go cue (target) and next vertical black dash line shows stop signal (non-target). The horizontal red areas show the significant (*p* < 0.01) between successful stop versus unsuccessful stop (response inhibition-related area) with Wilcoxon signed rank test. Sky blue indicates the significant area between successful go versus successful stop. Yellow indicates the significant area between successful go versus unsuccessful stop using Wilcoxon signed rank test.

**Figure 5 brainsci-10-00640-f005:**
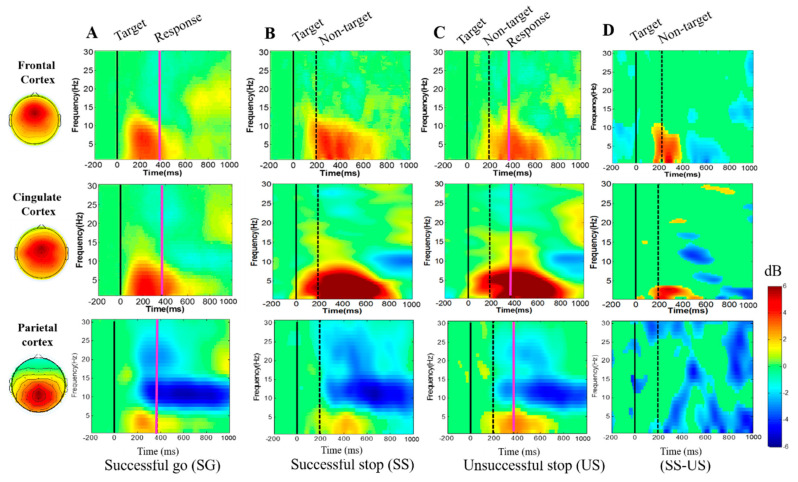
Averaged ERSP of all subjects in the frontal cortex, cingulate cortex and parietal cortex under (**A**) successful go, (**B**) successful stop, (**C**) unsuccessful stop, and (**D**) successful stop vs. unsuccessful stop. In the successful go condition, the black line indicates the go cue (target) stimulus; the pink line indicates response time. In the successful stop conditions, the dashed black line indicates stop-signal (non-target). Right color bar shows the ERSP in decibels (dB); statistical threshold at *p* < 0.05.

**Figure 6 brainsci-10-00640-f006:**
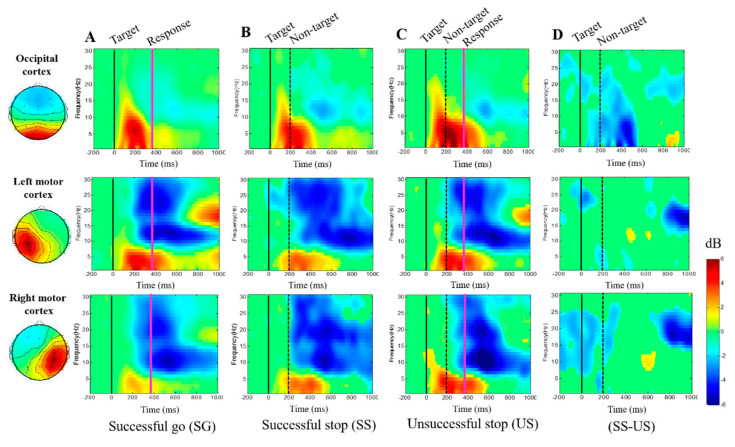
Averaged ERSP of all subjects in the occipital cortex, left motor cortex and right motor cortex under (**A**) successful go, (**B**) successful stop, (**C**) unsuccessful stop, and (**D**) successful stop vs. unsuccessful stop. In the successful go condition, the black line indicates the go cue (target) stimulus; the pink line indicates response time. In the successful stop conditions, the dashed black line indicates stop-signal (non-target). Right color bar shows the ERSP in decibels (dB); statistical threshold at *p* < 0.05.

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
