# Peer review of "Global Neural Activities Changes under Human Inhibitory Control Using Translational Scenario"

_brainsci, 2020, doi:10.3390/brainsci10090640_

Round 1
Reviewer 1 Report
This work used a stop-signal paradigm with a realistic battlefield scenario to examine changes in neural responses during inhibitory control. The authors used ICA techniques to identify brain regions contributing to neural responses to the visual stimuli in this task. N2 and P3 ERPs and associated ERSPs were examined in six component clusters, including frontal, central, parietal, occipital, left motor, and right motor brain regions. As described in more detail below, the clarity of current manuscript would be improved through a more explicit description of study goals and hypotheses as well as a discussion about how this work specifically adds to previous literature in the field. In addition, several concerns were raised regarding descriptions of the methods and results within the manuscript text as well as within the provided figures, as outlined in more detail below.
General Comments:
Introduction:
- As it is currently written, the introduction does not provide the reader with a clear motivation for the study, does not clearly describe the research questions or hypotheses, and does not clearly state how this study adds to the current literature and knowledge in this topic area. For example, the authors mention that the battlefield scenario represents a “new experimental design,” but a study using this design was completed and published by the authors in 2016. The authors also mention that this work represents the first time that ICA is used in this line of study, but it is again not clear how the ICA used in the current work differs from the ICA procedures used by Ko et al. (2016). The current paper would be improved if the authors could summarize this previous work in the context of the current study. The authors should revise the introduction to more clearly justify the study aims, and should also explicitly provide their main research questions and hypotheses.
Methods:
- The authors should provide a more detailed description of how the six component clusters were chosen. In addition, it would be useful to know how many participants and how many independent components contributed to each of the six clusters. Furthermore, is it possible to provide estimates of how much ERP variance each cluster explains?
The authors should also include details about what the ERSP measure is and how it was calculated in the current study, including some text about how the difference between ERSP in the successful and unsuccessful stop conditions was calculated and how results from this particular measure should be interpreted.
- The authors mention that the data was epoched “-200 ms to -1 ms before the stimulus, and 1 ms to 1000 ms after the stimulus.” Can the authors provide justification for why this was done, compared to epoching the data from -200 ms to 1000 ms?
- In addition, the authors should mention in the methods section whether 0 ms refers to the onset of the “go” stimulus or the onset of the “stop” stimulus as well as provide justification for their choice.
It seems likely that both the “go” and “stop” stimuli would elicit separate neural responses time-locked to the onset of each stimulus when both are used within a stop trial. It would be helpful if the authors could provide a discussion about this potential issue, including how these overlapping neural responses to both stimuli presented within the stop trials may or may not impact the current results and how differences in SSDs across trials might further complicate the issue.
Furthermore, based on the description of the test paradigm, the delay between the “go” and “stop” signals varied to change the difficulty of the task. Is the vertical line depicting the onset of the “stop” stimulus in figures 4-6 the average SSD (reported as ~194 ms) across all participants and all stop trials? It would be helpful if the authors could provide a clear explanation of this in the text as well as a discussion about how this might influence results. For example, if the ERP waveforms for each component are plotted and analyzed in reference to the onset of the “go” stimulus, is it possible that the increased response latencies observed in the “unsuccessful stop” trials (for example, for the N2 and P3 peaks plotted in Figure 4, panels A and B) are at least partially explained by the fact that the onsets of the stop stimuli in these “unsuccessful stop” trials are likely to occur later in time compared to the onsets of the stop stimuli in the “successful stop” trials, as SSD was increased to make the task more difficult?
Results:
- Up to this point, the paper has only discussed the relevance of N2 and P3 ERP peaks, but Figure 4 and the corresponding manuscript text includes the identification and discussion of P1, N1, and P2 peaks. It would be helpful to the reader if the authors could describe these peaks and justify their significance in the context of the current research questions.
- Based on the waveforms provided in Figure 4, it appears as if the timing of each peak is relative to the onset of the “go” stimulus. However, the written descriptions provided in the text imply that these time points may be relative to the onset of the non-target “stop” stimulus. (For example, “…the significant N2 negative wave around 400 ms after non-target…”). The clarity of the manuscript would be improved if the authors could provide a clearer and more consistent description of how the ERP peaks relate to the onset of the “go” stimulus and the onset of the “stop” stimulus across each condition.
In addition, the text should be revised to more clearly describe differences between each ERP waveform. For example, the text currently states that, “…we observed a significant P3 positive wave…in the frontal and central cortices of the brain under successful stop than in unsuccessful stop and go trials.” Visual inspection of panels A and B in Figure 4 suggests that N2 and P3 peaks are still present in the unsuccessful stop conditions, but that they occur at longer latencies and possibly smaller amplitudes compared to the successful stop conditions. The text should be revised to make these differences clearer for the reader.
- The labeling of N1, P1, and P2 peaks in Figure 4 do not seem accurate. For example, N1 is labeled as a positive going peak in Panels A and B which is then followed by the negative going N2 peak. Should this positive going peak instead be labeled as P2? In addition, this same positive-going peak in Panels C and D is labeled as P1 instead of P2. Figure 4 as well as the text should be revised for accuracy and clarity.
- The text referencing the data displayed in Figures 5-6 is also hard to follow. It would be helpful if the authors could revise the text and figure labels so that they are consistent. The text also mentions a “black colored rectangle” which does not appear to be included in the figures.
Minor Comments:
- Was a specific time window chosen for ICA analysis?
- The manuscript would be improved if the authors’ current description of how each behavioral measure was analyzed was moved to the methods section, as opposed to being placed in the results section of the paper.
- The individual data points as well as the symbols representing the average dipole sources are very difficult to see in each panel of Figure 3. The authors should revise this figure to make these data points easier to visualize and interpret.
- The clarity of Figure 4 is impacted by the blue and yellow highlighted regions overlaid onto the ERP waveforms, which make it difficult to identify between the line styles for the successful and unsuccessful stop conditions. Would it be possible to instead provide different colored asterisks that signify significant differences between conditions, similar to what was done for comparison between the stop conditions in red?
- Figure 5 should also be revised so that the line styles match across panels. For instance, column A uses the same dashed line to indicate participant responses as column B uses to indicate the onset of the non-target “stop” stimulus. Additionally, are the last panels in columns B and C of Figure 5 misplaced?
Author Response
Dear Editor,
We have revised the all statements of the manuscript carefully, according to reviewer suggestions. The authors are grateful to the editor and reviewer for the constructive comments on this manuscript.

Reviewer 2 Report
In their manuscript “Global Neural Activities Changes under Human Inhibitory Control using Translational Scenario” the authors study the human inhibition control in a realistic battlefield set up using EEG. The subjects get either real-life Go or NoGo signals from the game to shoot or not to shoot the enemy. The EEG analysis shows evidence for especially frontal and cingulate cortex engagement during Go-NoGO decisions. The results are consistent with previous literature.
The work demonstrates nicely how inhibitory control can be measured in a controlled way in a lab set up, still bringing more realistic features to the experiment. However, the text would benefit from serious editing and proof reading. Furthermore, some aspects of the study are not sufficiently explained to allow replication of the work. Also, the novelty of this work is overemphasized, because the last author already has a highly related work published in 2016.
- Please tone down the novelty value of this work! This is my major comment as this has also ethical issues. It seems that the authors emphasize the novelty of this work several times simply to appeal to the publisher.
Examples: “ Therefore, we designed a novel battlefield scenario to investigate the neural activities of inhibition in a more realistic environmental setting” How is this different from Ko et al. 2016?.
“To best our knowledge for the first time, we used independent component analysis with scalp maps and dipole sources’ locations to explore the neural activities under human inhibitory control in a realistic battlefield scenario.” How is this substantially different from Ko et al. 2016?
In the discussion, the authors MUST clearly compare their current results with Ko et al. 2016 and explain comprehensively, what is the true novelty of the presented work.
Ko, Li-Wei, et al. "Neural mechanisms of inhibitory response in a battlefield scenario: a simultaneous fMRI-EEG study." Frontiers in human neuroscience 10 (2016): 185.
- Please proof read the text more carefully. I recommend a professional to edit the text. For instance, “moreover” is over-used and oftentimes in such locations where it doesn’t logically fit. The sentence “The EEG has been a more reliable technique to investigate the neural activities of response inhibition at the millisecond scale [17].” doesn’t make any sense because EEG is not compared to anything. Instead of verb “firing” the authors have written “fairing”.
- The ERP signals from parietal cortex and motor cortex, and therefore any effect sizes are extremely small. Please discuss this and acknowledge this when drawing conclusions from the results.
- In the discussion, the authors describe how well the obtained results correspond to the traditional Go No-Go results. What is then the true novelty of this battlefield setup, if the results are largely the same? Or does this suggest that in fact, the conventional methods have been sufficient for capturing the cortical mechanisms behind inhibitory control?
- Things needing clarification:
- I simply cannot understand how the SSRT measure could work (SSRT=GO-RT-SSD). It doesn’t quantify in anyway the internal processes related to stopping.
- In chapter 4.1. N2 is said to refer to 400 and P3 to 500 ms. As far as I understood, they refer to 200 and 300 ms respectively, please clarify.
- Were the sounds on, while the participants played the game?
- How was the exact timing of the game events integrated with the EEG recordings?
Author Response

(The authors gave the same response as above.)

Round 2
Reviewer 1 Report
Several of the authors’ responses to comments from the first review as well as the authors’ subsequent revisions to the text are not adequate. In addition, many comments from the first revision have not been addressed. Several of the points that were not clearly addressed are those that may reflect major issues with the interpretation of the current results. These comments are reiterated below.
- The authors state that the purpose of the current study is to verify that the battlefield scenario paradigm is valid for a larger sample size (20 subjects vs. 11 previous subjects). I agree that it is valuable to confirm previous findings, but the authors failed to adequately revise the introduction to make this point clear to the reader, as previously requested. As it is currently written, the authors still claim that the current study is the “first time” that this type of work is done, and a clear description of and comparison to the previous Ko et al. (2016) paper has not been added to the text in the introduction or discussion sections.
- The authors confirm that both the “go” and “stop” stimuli contained within the stop trials each elicited neural responses but then say that they “did not use any overlapping EEG signals in this study”. This explanation is not clear. The use of separate event markers will not allow for a separation of overlapping neural responses to the two visual stimuli presented during the stop trials. See Kok et al. (2004) for a discussion of this issue.
- Since the ERP waveforms for each component are plotted and analyzed in reference to the onset of the “go” stimulus instead of the onset of the “stop” stimulus, it seems possible that the increased response latencies observed in the “unsuccessful stop” trials (for example, for the N2 and P3 peaks plotted in Figure 4, panels A and B) are at least partially explained by the fact that the onsets of the stop stimuli in these “unsuccessful stop” trials are likely to occur later in time compared to the onsets of the stop stimuli in the “successful stop” trials, since it was reported that the time between the go and stop signals was systematically varied based to make the task more or less difficult.
- While the amplitude of the peak labeled as “N1” in Figure 4 A&B does have a negative amplitude value, it is a positive-going deflection, and would therefore conventionally be labeled as a positive peak, not a negative peak. See examples in Figure 3 of Kok et al. (2004) and Figure 1 of Woodman (2010):
Kok, A.; Ramautar, J.; de Ruiter, M.; Band, G.P.H.; Ridderinkhof, K.R. (2004). ERP components associated with successful and unsuccessful inhibition in a stop-signal task. Psychophysiology, 41(1), 9–20.
Woodman, G. F. (2010). A brief introduction to the use of event-related potentials in studies of perception and attention. Attention, Perception, & Psychophysics, 72(8), 2031-2046.
- Issues with the clarity of the figures should be taken seriously. While the labels added to the first panel of Figure 3 point out the average and single dipole sources, this addition does not fix the fact that the data points, especially those representing the average dipole sources, are very difficult for the reader to see in each panel. In addition, Figure 4 was not revised, as the authors stated that it was too time consuming. Furthermore, the Figures 5 &6 or the text referencing the data displayed in Figures 5-6 has not been revised. These figures do not contain I, II, or III as labels for the rows, which are referenced in the text.
- The in-text written descriptions of the ERP waveforms shown in Figure 4 still describe the latency of each peak relative to the “non-target (stop signal)” instead of the target/go signal.
Author Response
Dear Editor,
We have revised the all statements of the manuscript carefully, according to reviewer suggestions again. The authors are grateful to the editor and reviewer for the constructive comments on this manuscript.We do our best to revise the manuscript. Many Thanks
